# Urbanisation generates multiple trait syndromes for terrestrial animal taxa worldwide

Cities can host significant biological diversity. Yet, urbanisation leads to the loss of habitats, species, and functional groups. Understanding how multiple taxa respond to urbanisation globally is essential to promote and conserve biodiversity in cities. Using a dataset encompassing six terrestrial faunal taxa (amphibians, bats, bees, birds, carabid beetles and reptiles) across 379 cities on 6 continents, we show that urbanisation produces taxon-specific changes in trait composition, with traits related to reproductive strategy showing the strongest response. Our findings suggest that urbanisation results in four trait syndromes (mobile generalists, site specialists, central place foragers, and mobile specialists), with resources associated with reproduction and diet likely driving patterns in traits associated with mobility and body size. Functional diversity measures showed varied responses, leading to shifts in trait space likely driven by critical resource distribution and abundance, and taxon-specific trait syndromes. Maximising opportunities to support taxa with different urban trait syndromes should be pivotal in conservation and management programmes within and among cities. This will reduce the likelihood of biotic homogenisation and helps ensure that urban environments have the capacity to respond to future challenges. These actions are critical to reframe the role of cities in global biodiversity loss.

Cities across the globe host significant biological diversity[1,2] that provides key ecosystem services for over 50% of the world's human population[3]. Urban growth often coincides with regional and global biodiversity hotspots[4] and occurs fastest in low-elevation, biodiversity-rich coastal zones[5]. Thus, although urban environments cause significant loss and transformation of habitats and modify landscape spatial structure, minimising these impacts will be critical if we are to counter their role in the current extinction crisis[6]. Understanding how multiple taxa respond, through their functional traits, to the environmental pressures and filters of urbanisation globally is essential to formulate effective strategies to promote biodiversity in urban environments.

Although considerable progress has been made toward understanding the impacts of urbanisation on global biodiversity, certain key research gaps remain. The scientific literature is geographically biased towards larger metropolitan areas[7] of the Northern Hemisphere and Australia[5]. Meanwhile, most biodiversity hotspots are in the tropics and the Southern Hemisphere and have received less attention[8]. Urban landscape structure has largely been characterised by negative aspects such as the proportion of impermeable surfaces, whereas the enabling aspects for biodiversity such as spatial configuration and the proportion of vegetation cover are relatively understudied[9], especially at the global level. Urban biodiversity studies are also heavily biased taxonomically towards plants and birds[10]. Other speciose and functionally-important groups, such as insects, amphibians, bats, and reptiles are severely impacted by urbanisation but poorly studied[11–14]. Despite the increasing importance of functional traits in the ecological literature and recent efforts to integrate functional aspects of

✉ e-mail: amy.hahs@unimelb.edu.au; bertrand.fournier@uni-potsdam.de; johan.kotze@helsinki.fi; marco.moretti@wsl.ch

biodiversity into urban ecological research[15], most urban biodiversity investigations remain focused on taxonomic diversity[16]. This hampers our ability to develop a mechanistic understanding of the impact of urbanisation on biodiversity; creates additional challenges when making cross-taxon or cross-region comparisons[17]; and hinders our ability to effectively conserve species with different life histories and habitat requirements.

Traits are the attributes of a species that describe morphology, phenology, behaviour, and life history and influence all aspects of an organism's fitness[18]. Trait-based approaches characterise the functional aspects of biodiversity[19], facilitate cross-taxon and cross-region comparisons[20], and provide insights into the ecological processes driving species assemblages[21]. Trait-based approaches are particularly suited to investigating the drivers of local community composition, including environmental filtering and biotic interactions[22,23]. Such knowledge is critical to the understanding and proactive mitigation of the effects of urbanisation on biodiversity and its associated ecological functions.

Cities impose strong filters on local faunal assemblages ranging from habitat loss to changes in local climate and environmental conditions and novel habitats and species interactions[24]. This filtering process is hypothesised to lead to global biotic, taxonomic and functional homogenisation, such that well-adapted species with similar traits or life histories become increasingly widespread geographically and locally abundant[25–27]. Cosmopolitan generalist species are found in most cities around the world[1], while specialist species tend to disappear[28]. Although exceptions exist, cities tend to select for small and highly mobile fauna that have a broad environmental niche and a generalist diet[15,29,30]. While evidence for global functional homogenisation remains inconclusive due to different legacies and regional species pools, leading to the high variability of local biodiversity in cities[31], current understanding suggests that highly urbanised environments favour mobile and r-reproductive strategist species with a generalist diet, leading to a decrease in functional diversity. We hypothesise that increased representation of these traits across multiple taxa in cities around the world supports the proposition that there is an 'urban syndrome' associated with species' responses to urbanisation[27]. This study sets out to:

1. Test our hypothesis by evaluating evidence against the current understanding of an 'urban syndrome' related to average community traits and/or functional diversity;
2. Investigate whether the proportion and spatial aggregation of urban land and forest cover (see Methods) induce stronger changes in community functional diversity than known latitudinal or climatic trends. In this case, we use urban land cover to represent a gradient of urbanisation filters, and forest cover to represent the amount of tree canopy cover;
3. Investigate the spatial scale at which the proportion of urban land has the strongest effect, and how this differs among functional groups.

This study used a collaboratively compiled dataset of 5302 species found in >70000 plots across 379 cities from 48 countries (Fig. 1) to investigate how urbanisation shapes the community trait-composition and diversity of six terrestrial animal taxonomic groups (amphibians, bats, bees, birds, carabid beetles, and reptiles) across the globe. The data are a collation of empirical studies at the highly-resolved spatial scale of individual sites rather than generalised to city. Only one taxon (birds) was extracted from a global biodiversity dataset (eBird). We acknowledge there are still geographic biases in the data that reflect legacies of studies published prior to 2017[10]. We are also aware that there are additional taxa groups that we would have liked to include but lacked the capacity to consider in this project. However, to our knowledge, this is the most comprehensive compilation to date of urban biodiversity data for several terrestrial animal taxa at the site scale. The six taxa represent a broad range of natural histories, ecologies and behaviours and have sufficient occurrence data and trait information to conduct a global study, despite some geographic biases. The traits we considered were body size, diet, mobility and reproductive strategy, as these are all important for an individuals' survival, growth and reproduction[18]. Functional diversity metrics captured key facets of trait diversity (functional richness – FRic, functional evenness – FEve, functional dispersion – FDis), to investigate whether there was evidence to support a contraction of trait space associated with the urban syndrome. Further details can be found in the Methods.

Our global analysis shows that urbanisation is a major driver of urban community functional composition and identified four general

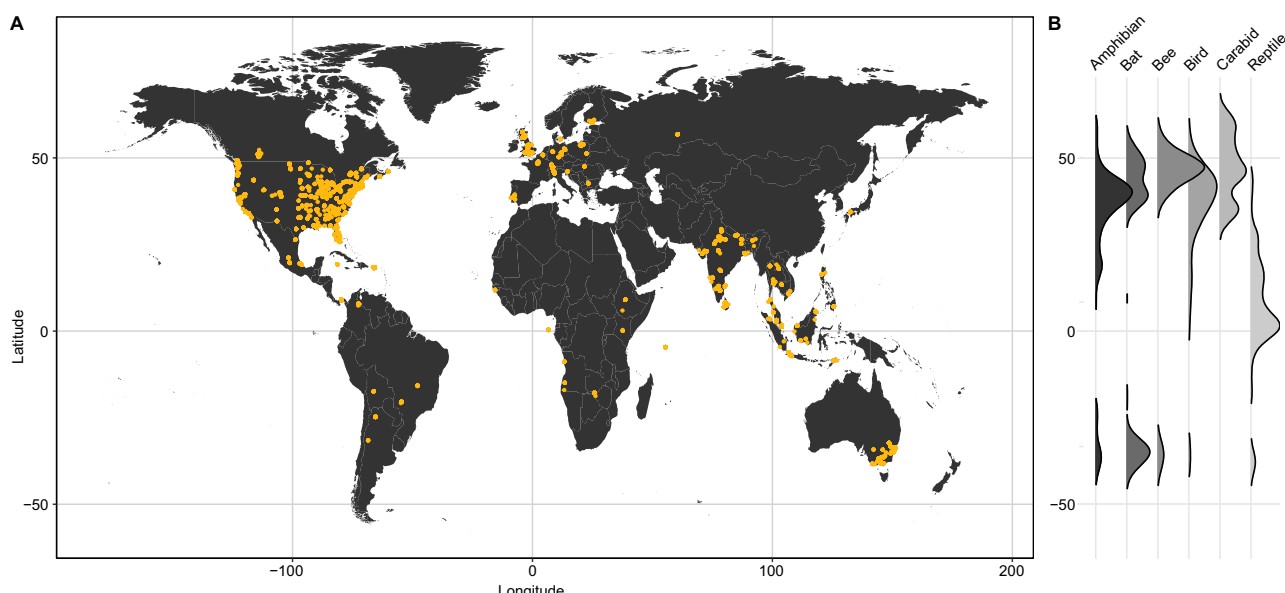

**Fig. 1 | Global distribution of data included in this study. A** Locations of sampling plots (orange dots) for all six taxonomic groups combined. All data are from the UrBioNet contributor network except for birds (eBird). **B** Ridgeline plots show the density of sampling locations per taxon as a function of latitude. See Supplementary Fig. 1 for taxon-specific maps.

urban trait syndromes (mobile generalists, site specialists, central place foragers, and mobile specialists). These diverse urban trait syndromes may have important consequences for the management and conservation of the biodiversity and functioning of urban ecosystems.

## Results

All traits and functional diversity metrics changed with increasing urban land cover, although the strength and direction of change within each trait category differed among taxa (Fig. 2).

Body size and mobility were affected differently by urbanisation depending on the taxa (Fig. 2). Carabids, birds and reptiles tended to have smaller body size (7%, 23% and 27%, respectively) in the most urbanised areas relative to the least urbanised areas. The mobility of carabid beetles was higher (19%), while that of reptiles and birds was lower (1%, 5%) in more urbanised areas. Amphibian and bat body size was larger (4%, 1%) in the most urbanised areas. Amphibians were 4% lower in mobility, while the mobility of bats was slightly higher (1%) (Fig. 2). For bees, inter-tegula distance is the trait most frequently used to represent body size and mobility, and showed an inverted u-shape, where the linear trend showed a slight increase (< 1%).

Our results suggest that increased urban land cover can induce a shift toward a more specialist or generalist diet depending on the taxa considered (Fig. 2). Specifically, omnivory was favoured in areas with the highest urban land cover for birds (19%) and carabid beetles (14%). Bees showed a u-shaped response with a linear trend towards a 3% larger proportion of short-tongued species (Fig. 2). However, amphibians and reptiles had more specialised diets in areas with the highest urban land cover (8% and 5% respectively).

Reproductive traits were the first (bats and carabids) or second (amphibians, bees, birds and reptiles) most affected trait when considered across all traits for a taxon (Fig. 3). The reproductive strategy trait had the highest proportion of variance explained for four taxa, explaining 48 – 65% of the variance for bats, bees, carabids and reptiles ($R^2_{obs-pred}$ in Table 1). The exceptions were amphibians and birds where feeding or body size (respectively) were more important. The statistical trends indicated that larger urban land cover was associated with reduced clutch size (amphibians, birds and reptiles), more generalist roosting (bats), overwintering as imago (carabids) and solitary nesting (bees) (Fig. 2). Bats with generalist roosting requirements increased by 3%, bees that were solitary nesters increased by 9% compared to social nesters, and carabids showed a 4% increase in the proportion of species that overwinter as adults.

The effect of urban land cover was most important at the largest spatial scale considered for all taxa examined (1000 m for birds, 500 m for all other taxa; Fig. 4). The importance of the proportion and spatial aggregation of urban land cover as predictors of taxon-specific trait syndromes ranged from 3% to 20% depending on the taxon (light blue bars, Fig. 4), but composition (%) was consistently stronger than arrangement (agg). Metrics related to forest cover were generally the least important across all taxa, with birds being the exception. Latitude and climatic region predicted shifts in community functional composition of most taxa better than urban or forest land cover or configuration. The only exception to this was again for birds, for which the importance of latitude was equal to the importance of forest cover (%) within 1000 m of the site.

There were clear effects of urbanisation on all facets of functional diversity and species richness, however, they varied between taxa. Functional richness (FRic) was the functional diversity facet that was best predicted by the extent and aggregation of urban land cover for amphibians, bees, carabids and reptiles (Fig. 3; $R^2_{obs-pred}$ in Table 1) but the direction of the response varied (Fig. 2). Along the urbanisation gradient, functional richness (FRic) for bats (6%) and reptiles (9%), showed a u-shaped response for amphibians and birds and tended to increase in bees (2%) and carabids (8%) (Fig. 2). Functional dispersion (FDis) was a more important dimension of functional

diversity for bats and birds (Fig. 3, $R^2_{obs-pred}$ in Table 1), which was 4% (bats) or 5% (birds) lower in areas with the highest urban land cover. Functional evenness (FEve), although overall poorly predicted by our models, was the dimension of functional diversity that most consistently responded strongly to urbanisation (% MSE in Table 1). Like functional diversity dimensions, species richness showed different statistical trends depending on the taxon considered. The species richness of carabid beetles and reptiles was higher (1% and 2%, respectively) in areas with the highest urban land cover but the richness of all other taxa was lower (3% amphibians, 8% bats, 18% bees, 17% birds; Fig. 2).

## Discussion

We have shown that urbanisation produces taxon-specific changes in trait composition, with traits related to reproductive strategy consistently showing the strongest response. Our results suggest that the effect of urbanisation on functional traits results in four urban syndromes. This finding contrasts with the view that there is a single global 'urban trait syndrome' associated with species' responses to urbanisation and has far-reaching implications for a better understanding of ecological community dynamics and biotic homogenisation in urban ecosystems.

Body size and mobility are frequently correlated in functional trait studies: larger species tend to be more mobile[32]. Mobility is likely to be favoured when it helps an organism acquire resources and/or avoid competition and predation. However, our results show that for some terrestrial animal taxa (e.g. amphibians and reptiles, Fig. 2), urbanisation may select for species with small home ranges that can exploit local resources[33] and avoid risks associated with the urban matrix[34]. Reduced mobility in these taxa make them particularly vulnerable to habitat loss or degradation and can lead to the isolation of populations, increasing the importance of genetic drift and local population extinction risks.

Increasing proportions of omnivores with increasing urban land cover was observed for birds (19%) and carabid beetles (14%), which aligns with a common finding that dietary breadth predicts success in urban environments[35,36], and our hypothesis for an 'urban syndrome'. Bees showed a u-shaped response, which may reflect a wider diversity of flowering plants being available in urban areas, thereby providing a variety of resources for both generalist and specialist feeders. Amphibians and reptiles showed shifts towards species with increasing dietary specialisation. This specialisation may enable finer niche partitioning in spatially constrained spaces and thereby avoid some of the impacts of urban environments through more efficient foraging[37]. Overall, our results highlight that both generalist and specialist feeding strategies can be selected for in urban environments but will depend on the interplay between the composition and distribution of food resources and the species ability to access and utilise them.

Our results provide evidence that urbanisation strongly selects for species with the capacity to find suitable conditions for reproduction. Fewer suitable nesting sites and higher risk of disturbance/predation in cities can thus have a strong impact on community functional composition. Providing supplemental nesting resources to compensate for loss of natural nesting possibilities can limit this impact, as has been demonstrated by the use of nest boxes to supplement the loss of hollows[38]. Increased urbanisation also influenced community mean clutch size. For example, reptiles clutch size was 27% lower, while birds displayed u-shape negative trend with 7% variation in clutch size (Fig. 2). A previous global analysis found that reptiles tend to have larger clutch sizes at higher latitudes where suitable conditions for breeding are constrained by short growing seasons or other limitations that select for reproductive strategies that maximise the number of offspring produced when food availability peaks[39]. In cities, the reduction in frost days due to the urban heat island and the greater consistency of food and water throughout the year due to horticultural

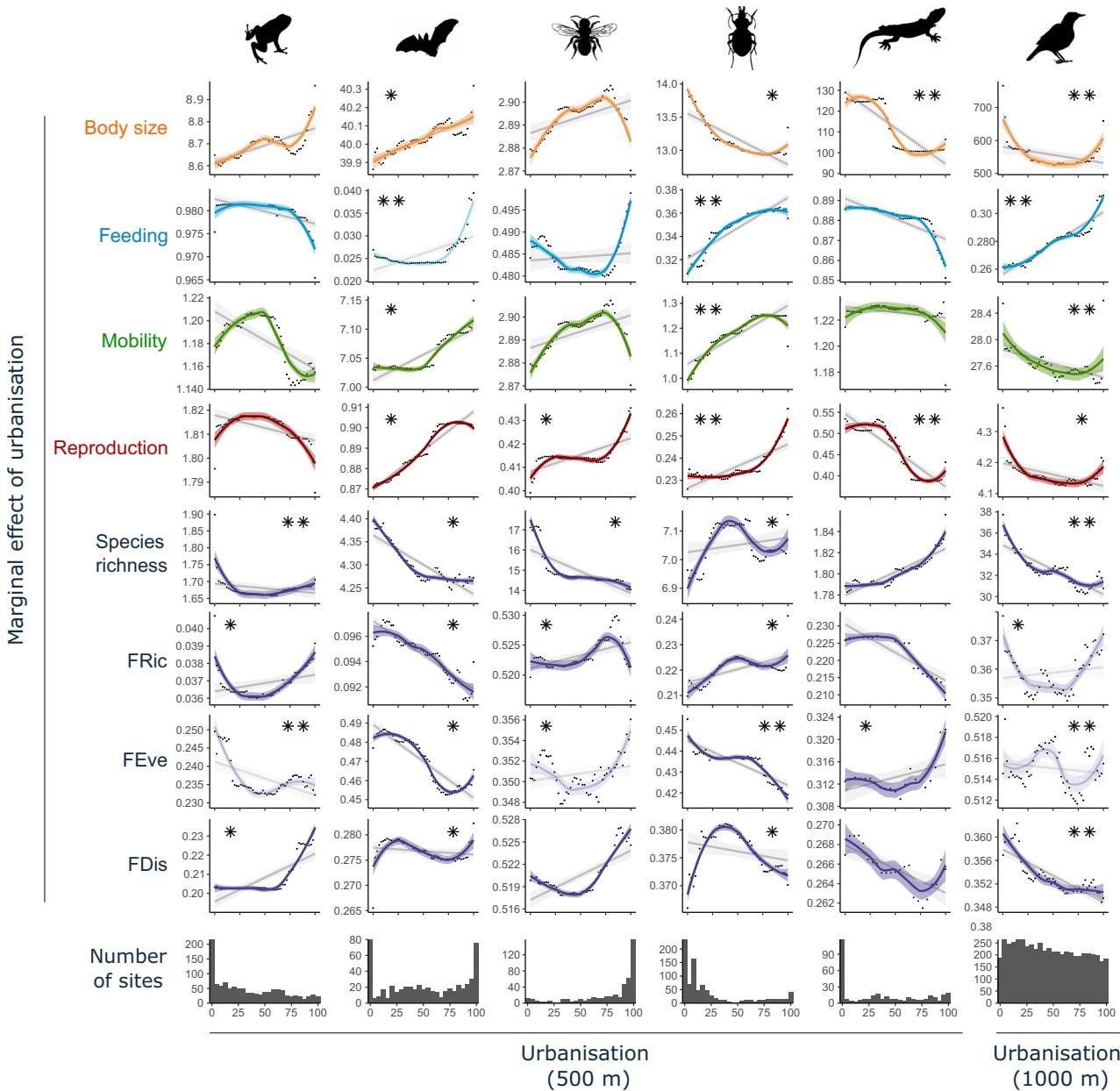

**Fig. 2 | Predicted changes in trait values per taxon along an urbanisation gradient.** Partial dependence plots showing the urbanisation-induced shifts in community functional metrics for six taxonomic groups. The partial dependence plots summarise the marginal effect that urbanisation (x-axis = percentage of urbanised area in a 500 m radius around the sampling plot; or 1000 m for birds) has on the predicted values of each community-level trait (i.e. effect of urbanisation when climate, latitude and forest cover are kept constant). The y-axes reflect the range of predicted values for each response variable (community-weighted mean trait values and diversity metrics) and are not zeroed so care should be taken when interpreting the magnitude of change. The fitted colour lines and 95% confidence bands around predicted values are from Local Polynomial Regression (LOESS). The grey lines and 95% confidence bands around predicted values (light grey) are from linear regressions based on the same data to indicate direction of trend. Trait definitions are provided in Supplementary Table 3 (briefly, Feeding: high values = generalist diet except for bats where feeding represents different hunting strategy; Mobility: high values = higher mobility; Reproduction: amphibians, birds and reptile = clutch size / other taxa = reproduction strategy). Note that for bees, the inter-tegula distance was used for body size and mobility, and therefore the model presented is the same for both traits. Functional dispersion (FDis), functional richness (FRic) and functional evenness (FEve) are defined in the method section in "Functional composition of animal communities" (see also Supplementary Fig. 2). Transparent shade represents models with <10% variance explained. Stars show the contribution of urbanisation to the overall model (* 20-50%; ** > 50%). Additional information on each models' overall predictive power and the contribution of the percentage of urban land cover can be found in Table 1. Image credits: Ghedo and T. Michael Keesey (https://creativecommons.org/licenses/by-sa/3.0/) for reptile. Michael Keesey (vectorization); Thorsten Assmann, Jörn Buse, Claudia Drees, Ariel-Leib-Leonid Friedman, Tal Levanony, Andrea Matern, Anika Timm, and David W. Wrase (photography) (https://creativecommons.org/licenses/by/3.0) for carabid beetle. All other silhouette images come from www.phylopic.org and are public domain images.

plantings and human activities, may benefit species that have multiple but smaller clutches to avoid population density pressures on locally limited resources. Smaller clutch sizes in urban birds have been associated with higher survival and increased growth[40]. Reduced clutch sizes in birds have also been linked to perceptions of increased predation risk in altricial species where the young are fed and protected by parents when they are first born[41]. Future research could look more closely to understand to what extent the shift in clutch size represents

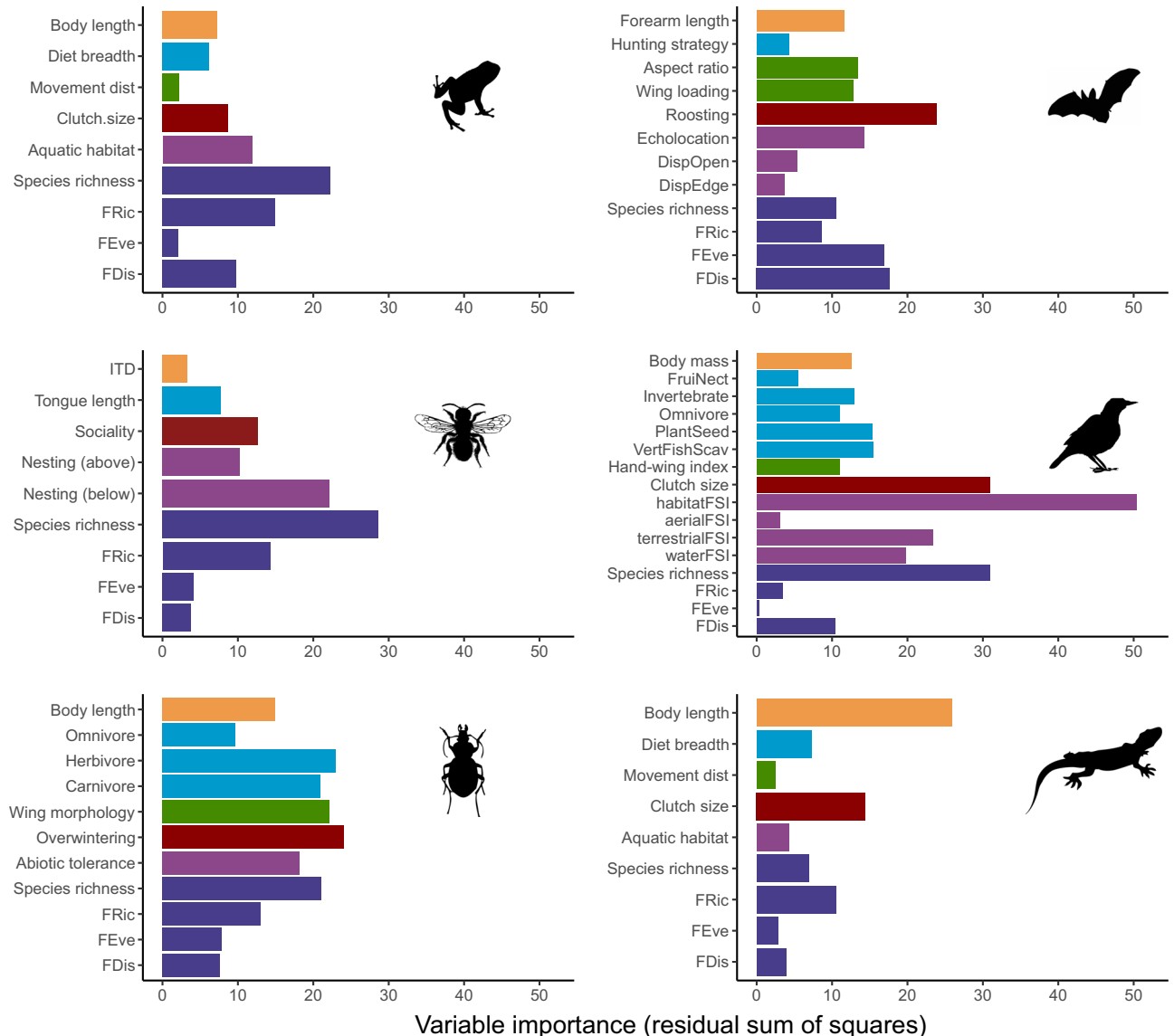

**Fig. 3 | Relative importance of the extent and aggregation of urban land cover as predictors of community means (colours show the different trait categories; Supplementary Table 3) and variability (FDis = functional dispersion, FRic = functional richness, and FEve = functional evenness; dark blue) of traits as well as species richness for each taxonomic group.** Variable importance was estimated using the residual sum of squares from random forests models. Average variable importance values weighted by the R² of the test set of each individual model were computed to estimate urban land cover variable importance for each metric of community-weighted means and variability of traits. Longer bars indicate traits or functional diversity measures that are better predicted by urban land cover within the surrounding landscape. Image credits: Idem Fig. 2.

a change in the number of species exhibiting a given development type as altricial birds have smaller clutch sizes than precocial birds that require little parental care[42].

Our results confirm the effect of latitude and climate as key drivers of the functional biodiversity of taxa observed in cities. Landcover effects were strongest at the largest spatial scales considered (1000 m for birds, 500 m for all other taxa), and the composition of the landscape (% cover) was more important than configuration (agg). These results highlight the importance of landscape-level management of urban biodiversity and the role of spatial context. They also provide additional support for our proposed general urban trait syndromes, which are highly influenced by the distribution and abundance of resources within the landscape.

We acknowledge that processes occurring at larger spatial scales than those considered in this study can also be important, especially for species with high mobility. Equally, there may also be finer-scale processes that we were not able to consider due to the resolution of available datasets. Future research could address these limitations or could expand our approach to look at a wider range of taxa. The study could also be repeated in the future when empirical data from a wider range of geographic regions are available to test how well the patterns observed here continue to apply.

**Four general urban trait syndromes, rather than one universal syndrome**

Our study indicates that rather than a single urban syndrome, there is strong evidence to support that each taxon has an individual urban trait syndrome which can be classified into one of three typologies: mobile generalists, site specialists, or central place foragers (Fig. 5), and could hypothetically include a fourth typology: mobile specialists. The urban trait syndrome for mobile generalists most closely matches our original hypothesis that urbanisation selects highly mobile species with more generalist diets and reproductive strategies that are better able to exploit available resources. This syndrome was observed in

**Table 1 | Performance of models predicting traits and diversity metrics**

|  |  | Amphibians | Bats | Bees | Birds | Carabids | Reptiles |
|---|---|---|---|---|---|---|---|
| Body size | $R^2_{obs\text{-}pred}$ | 62 | 44 | 32 | 18 | 40 | 62 |
|  | % inc MSE | 17 | 25 | 12 | 62 | 47 | 62 |
| Feeding | $R^2_{obs\text{-}pred}$ | 67 | 9 | 55 | 20 | 19 | 55 |
|  | % inc MSE | 16 | 52 | 13 | 57 | 83 | 13 |
| Mobility | $R^2_{obs\text{-}pred}$ | 17 | 31 | 32 | 16 | 46 | 42 |
|  | % inc MSE | 14 | 42 | 12 | 62 | 71 | 8 |
| Reproduction | $R^2_{obs\text{-}pred}$ | 62 | 65 | 57 | 52 | 48 | 33 |
|  | % inc MSE | 19 | 40 | 38 | 44 | 57 | 93 |
| Sp. Richness | $R^2_{obs\text{-}pred}$ | 68 | 56 | 80 | 46 | 61 | 70 |
|  | % inc MSE | 53 | 29 | 48 | 70 | 39 | 15 |
| FDis | $R^2_{obs\text{-}pred}$ | 53 | 54 | 26 | 16 | 18 | 29 |
|  | % inc MSE | 31 | 34 | 13 | 50 | 49 | 15 |
| FRic | $R^2_{obs\text{-}pred}$ | 53 | 29 | 46 | 20 | 59 | 59 |
|  | % inc MSE | 39 | 23 | 36 | 38 | 40 | 18 |
| FEve | $R^2_{obs\text{-}pred}$ | 5 | 50 | 10 | 8 | 17 | 11 |
|  | % inc MSE | 60 | 48 | 30 | 64 | 71 | 22 |

Summary statistics of random forests models of community-weighted means of traits and functional diversity metrics. "$R^2_{obs\text{-}pred}$" is the performance of the model on the independent test dataset where high values indicate that the response variable is well-predicted by urban and forest land cover, climate, and latitude. "$R^2_{obs\text{-}pred}$" was calculated as R-squared of the relationship between the predicted and the observed values of the independent test dataset. "% inc MSE" is the average increase in squared residuals when the variable is permuted. It represents the specific contribution (or importance) of the percentage of urban land cover (within a 500 m radius for all other taxa except birds for which we used a 1000 m radius) to the overall model performance. High values suggest that urban land cover is an important predictor.

bats and carabid beetles, with both groups displaying increases in traits related to mobility and generalist diets, a broader range of roosting sites for bats, and an increase in the proportion of species overwintering as adults in carabids. The shift in body size for these two taxa differed, but in ways that were consistent with increased mobility. Bats showed an increase in body size, which is consistent with previous studies that found urban environments tend to select larger bats that are stronger and more rapid fliers, and that forage on insects in open settings using echolocation[43]. Carabids displayed a shift towards smaller-bodied species[30] that can fly[44], a set of traits that enables greater mobility and an increased capacity to seek out food resources, without the need for strong site fidelity as observed in the central place forager or site specialist urban trait syndromes.

The urban trait syndrome associated with site specialists was characterised by reduced mobility, increased dietary specialism and a shift towards smaller clutch sizes. All these traits are advantageous to species that are reliant on highly localised life cycles either due to resource scarcity or increased risk of mortality in the urban matrix due to predation, pollution or vehicle collision. The taxa that displayed this urban trait syndrome were amphibians and reptiles. Dietary speciali-sation could allow multiple species to co-exist within a more con-strained physical space through resource partitioning, while reduced clutch sizes would help minimise density-dependent mortality in species that are not highly mobile. Alternatively, remnant urban green spaces could act as ecological traps that disproportionally affect spe-cialised species over generalist ones[45], with diversity eventually decreasing as the extinction debt becomes realised[46].

Central place foraging is an evolutionary ecology model that has been used to describe the foraging strategies for bees, mussels and other taxa[47]. As the name suggests, central place foragers establish a home base location from which they undertake daily movements to forage for additional resources. The taxa that displayed this urban trait syndrome in our study were bees and birds. Bees showed a shift towards a more solitary reproductive strategy, reduced mobility and increased dietary generalisation at very high levels of urbanisation (>80%, Fig. 2). For bees, this trait syndrome is consistent with pre-viously documented movements observed in urban systems[48]. For birds, this trait syndrome was associated with reduced mobility and

clutch sizes, similar to the site specialists discussed above, but accompanied here by an increase in the proportion of omnivory which would allow the individual to exploit a wider range of resources in the area surrounding their nest.

The final urban trait syndrome we propose would be associated with mobile specialists and is characterised by species that are able to meet their resource needs by being dietary specialists that are highly mobile and can move between spatially isolated food sources without having to return to a central place. While this urban trait syndrome was not observed in our study, there is anecdotal support for it at the species level. Wetland birds offer a useful example, where their dis-tribution is tightly linked to a specific resource (waterbodies), but they have the capacity to easily move between locations when resources fluctuate.

While the general urban trait syndromes identified in this study are relatively clear and well supported, the associated shifts in func-tional diversity metrics and species richness are less consistent (Fig. 2). This may be due to differences among taxa in relation to large-scale factors such as legacy effects that control how and to what extent regional diversity influences local diversity through species-pool effects[49]. Alternatively, if urbanisation selects for ecological strate-gies (or trait syndromes) that allow taxa to maximise the use of avail-able resources, then the implications for functional diversity and species richness will be emergent properties of the species and taxo-nomic responses to the specifics of the resources in question. Depending on the heterogeneity and availability of resources, trait selection may result in an increase or decrease in particular trait combinations (FRic), with different levels of clustering (FEve) and expansion or contraction of the trait space (FDis). This filtering can affect community dynamics and stability through modifications of species interactions and demography[50], and likely changes the capa-city of urban biodiversity to respond to climate change and other stressors.

Our study was interested in community-level trait characteristics at the taxon level. Therefore, it is quite possible that individual species within each taxon belong to different urban trait syndrome groups. For example, small insectivorous birds may display traits characteristic of site specialists, while non-hollow-dependent parrots could display

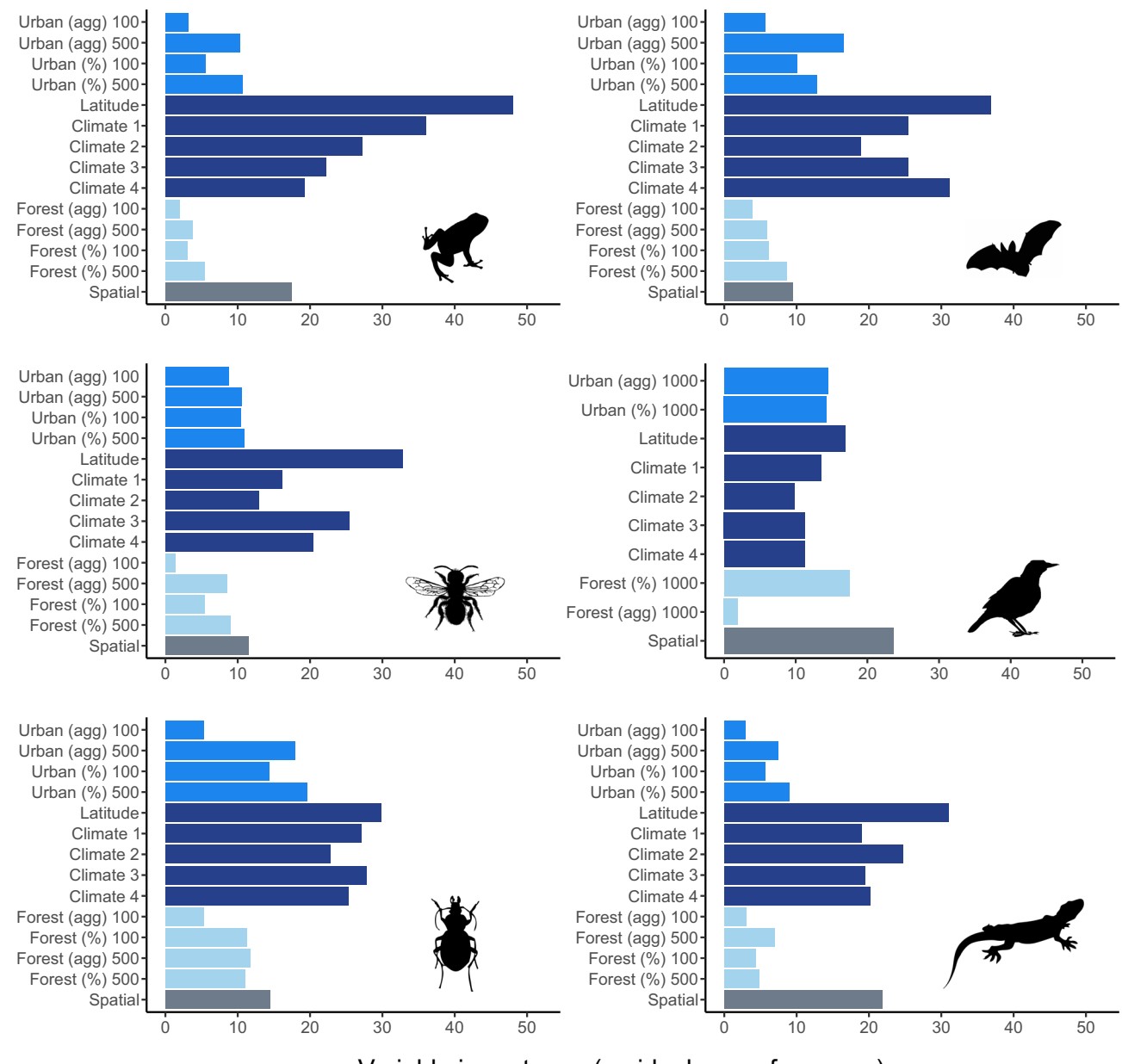

Variable importance (residual sum of squares)

**Fig. 4 | Relative importance of variables in predicting trait responses per taxon.** Importance of percent cover (%) and spatial aggregation (agg) of urban and forest land cover at different buffer distances (100 m and 500 m for most taxa; 1000 m for birds), latitude, climate PCA axes, and spatial covariates (dbMEM) as predictors of the trait syndrome (i.e. considering all community weighted means and functional diversity metrics) for each taxonomic group. Variable importance was estimated using the residual sum of squares from random forests models. Average values weighted by the $R^2$ of the test set of each individual model were computed to estimate variable importance for the overall trait syndromes. Image credits: Idem Fig. 2.

mobile generalist traits, and waterbirds could display mobile specialist traits. Similarly, bats are often considered to be central place foragers in other landscapes. Future research could investigate the degree to which these syndromes are representative of species within the different taxa, and how statistical trends in functional diversity emerge from species and taxonomic responses to resource availability in urban landscapes. This information could then be used to identify resources that are critically limiting for functional diversity in urban areas and guide actions aimed at making cities suitable environments for a wider range of species.

Our results provide further evidence to counter the fallacies that urban environments are biological deserts[2], and that biodiversity conservation is incompatible with urban areas[51]. Instead, they point to the importance of resources, particularly those related to

reproduction, as a critical filter in determining the diversity of terrestrial animals that persist in urban landscapes.

Since urbanisation occurs disproportionately in biodiversity hotspots[52], it has been framed as a strong driver of biodiversity loss at the global scale. Our analysis shows that the diversity of species (and functional traits) found within urban areas reflects the heterogeneity and availability of resources across the urban environment. Whether populations of site scale specialists are viable or small sites are acting as ecological traps will vary on a case-by-case basis, particularly when supportive human actions such as ecology with cities[53] are considered. Thus, our research presents a clear mandate to find innovative means of incorporating terrestrial animals' habitat requirements (particularly related to reproductive strategies) back into cities using both land-sharing and land-sparing approaches[54].

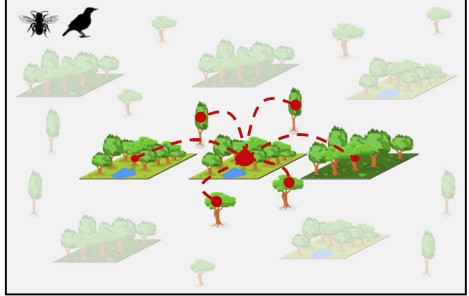

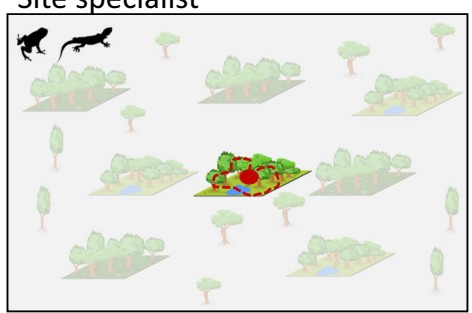

| Urban syndrome | Mobility | Feeding | Reproduction |
|---|---|---|---|
| Mobile generalist | Increased mobility | More generalist diet | Diverse strategies + diverse locations |
| Mobile specialist | Increased mobility | More specialist diet | Diverse strategies + reduced locations |
| Central place forager | Site fidelity | More generalist diet | Reduced strategies + reduced locations |
| Site specialist | Reduced mobility | More specialist diet | Reduced strategies + reduced locations |

**Fig. 5 | Simplified representation of the four urban trait syndromes.** Two types of green habitat patches with different resources are represented in an otherwise mostly unsuitable urban matrix. Grey patches represent green habitats that are unusable for a specific taxon. Red dashed lines show typical movement pattern of taxa among patches. The landscapes of Fig. 5 and their individual elements were created by Bertrand Fournier using the basic set of tools available in the vector graphics open source software "Inkscape 1.2.2". No previously-created elements were used. No images from a database were used. The landscapes of Fig. 5 and their individual elements were not published elsewhere. Image credits for taxa silhouettes: Idem Fig. 2.

To maximise urban biodiversity, conservation and management should identify those species most at risk of local extinction, then determine if there are options to incorporate any limiting resources back into the landscape. However, the complexity of responses and mechanisms observed in this study suggest that positive actions for one taxon (e.g., increasing tree canopy cover for birds) may disadvantage others (such as bees that forage in more open landscapes). It follows that identifying priorities in urban biodiversity management will become an increasingly important challenge that will need to be addressed at multiple spatial scales, across diverse taxa and sites, and using a systems approach. However, the fine scale heterogeneity present in urban landscapes and the call to provide a portfolio of places to cater to diverse human preferences both offer important signals that multiple resource needs can be met within the urban landscape.

Overall, our results suggest that resource distribution and abundance appear to filter taxa into one of four urban trait syndromes: mobile generalists, mobile specialists (nomads), central place foragers and site scale specialists. These urban trait syndromes can be applied at the level of individual species, but this study also suggests that predominant urban trait syndromes also emerge at the taxon level. Future research would be needed to untangle these relationships further. Accounting for diverse urban trait syndromes and integrating them into the planning, design and management of urban environments will become increasingly critical if we are to preserve diverse biotic communities essential to the functioning of urban ecosystems and reframe the role that cities play in the global biodiversity extinction crisis.

## Methods
### Urban biodiversity data
To identify potential datasets for our analysis, we conducted a systematic review of the published urban biodiversity literature from 1990 to 2016 to identify studies that met the following criteria: 1) community level data, 2) collected in multiple plots, and 3) across one or multiple cities. The Web of Science literature review was conducted using the following keywords:

TOPIC: (urban OR "peri-urban" OR periurban* OR suburban* OR "sub-urban*" OR conurbation OR city OR cities OR town* OR megalopol* OR metropol* OR "built-up" OR "built environm*")
AND
TOPIC: (Arthropod* OR Invertebrate* OR Insect*)
TOPIC: (Bats OR Chiroptera)

TOPIC: (Bees OR Hymenoptera OR Aculeata OR Pollinator*)

TOPIC: (Carabid* OR "ground beetle*" OR Cincidel* OR "tiger beetle*")

TOPIC: (Mammal* OR Marsupial*)

TOPIC: (Amphib* OR frog* OR toad* OR Anura OR Reptil*)

AND

Refined by: TOPIC: (Biodiversity OR "species composition" OR "species assembl*" OR "species communit*" OR "species richness" OR "species diversity" OR "Shannon-Weaver" OR "Shannon-Weiner" OR "Shannon-Wiener" OR "Shannon* H" OR "Simpson* index" OR "Simpson* diversity" OR "Simpson* dissimilarity" OR "Simpson* beta" OR evenness OR "alpha diversity" OR "beta diversity" Biodiversity OR composition OR assembl* OR communit* OR richness OR diversity OR Shannon OR Simpson OR evenness) AND TOPIC: (Gradient* OR Grid OR Site OR Map* OR Spatial* OR GIS OR Plot OR Sample OR Area OR Occurrence OR Presence OR distribution) AND WEB OF SCIENCE CATEGORIES: (ECOLOGY OR BIODIVERSITY CONSERVATION OR PLANT SCIENCES OR URBAN STUDIES OR FORESTRY)

Indexes=SCI-EXPANDED Timespan=1990-2016

From these studies, we identified those taxa with sufficient number of papers to permit a meta-analysis (e.g., in 2016 there were only 8 papers investigating snails in urban landscapes that met our criteria), and appropriate for the methods we proposed to use (e.g., fish are unlikely to be strongly influenced by % tree cover within 100 – 500 m).

Once we had finalised our taxa groups, we identified a core group of authors from the WoS literature review who had datasets that would be suitable for our proposed analysis, and contacted them to see if they would be interested in collaborating in this research. We also gained additional collaborators who volunteered their datasets after hearing our presentations on this project at the 2017 Ecological Society of America meeting, 2017 Ecological Society of Australia meeting and 2017 International Ecology Congress (INTECOL).

Researchers who responded to our invitation were provided with a data collection template to ensure we received all of the required information in a format that could readily be integrated into a larger dataset. This included a species x site table, species x trait table and site information table, and a link to a metadata form where we could capture additional information about the study used to produce the data.

To facilitate this process, we formed taxon-based groups which were coordinated by 2-3 members of the UrBioNet coordinating group. Throughout the project the taxa coordinators were responsible for collating the data into a single dataset, and worked with the data contributors to develop the compiled species x trait tables. Once the data had been compiled the coordinating group populated the standardised site information, analysed the data and created a preliminary results document that was circulated back to the data contributors for discussion and feedback within the taxonomic groups in March 2020. Feedback from this process was compiled back by the coordinating group and used to update the analysis. The coordinating group then drafted a manuscript that was shared back to the data contributors for feedback. After the feedback on the manuscript had been incorporated, a final version of the manuscript was circulated to ensure all named authors agreed to the submission of the manuscript. As the timing of the circulation of the initial round of results coincided with the onset of the COVID-19 global pandemic, the timeframes for delivering this project were disrupted, and some of the additional feedback rounds were bypassed as a large number of contributors had already met the criteria for authorship.

The exception to this process was the bird taxon, where we extracted data from existing global datasets to match the cities where we had information for other taxa groups.

As this research was hosted by the UrBioNet Research Coordination Network, we applied their Authorship Policy (https://sites.rutgers.edu/urbionet/about/authorship-guidelines/), where authorship required a substantial contribution beyond simply providing data or being present at the initial workshop. This is in alignment with Weltzin et al. (2006)[55] and other publications that seek to ensure authorship reflects a substantial contribution.

Our final dataset consisted of information from 72086 plots spread across 379 cities worldwide and retained six taxonomic groups with sufficient data for a global assessment of urbanisation effects (see Fig. 1, Supplementary Fig. 3, and Supplementary Tables 1 and 2): amphibians (140 species, 1202 plots in 191 cities), bats (84 species, 540 plots in 43 cities), bees (486 species, 471 plots in 25 cities), carabid beetles (327 species, 889 plots in 17 cities), reptiles (98 species, 321 plots in 71 cities) and birds (4167 species, 68558 plots in 177 cities). The latter was collected from the eBird global community-science program (https://ebird.org)[56], and covers the period from 1 January 2002 to 31 December 2018 from across the globe. We retained eBird checklists for analysis that were located within 1.5 km of the center of each city and were conducted using the P20, P21, P22, P23, P48, and P62 sampling protocols. We retained travelling surveys that were <1 km and area surveys that were <1 km2. We only considered observations that were identified as valid by the eBird review process, and we combined observations in grouped checklists into single checklists. While there are documented biases within this dataset[57,58], the signals are likely to be dampened in this study by including data points across a large number of globally distributed cities.

Within our study a plot is defined as an individual location where a survey was conducted. While we were unable to explicitly quantify a regional species pool for each taxon and city due to limitations of the available data, we were able to quantify the level of urbanisation in the surrounding landscape for each site and confirm that our data covered the full range of values (Fig. 2). Therefore, we are confident that our data include species outside the urban area and not simply species that are associated with urban environments.

For each taxon, we gathered functional trait data related to body size, diet, mobility and reproductive strategy, because these traits are important for an individuals' survival, growth and reproduction[18]. We deliberately included both native and introduced species as we were interested in understanding global trait responses of species, as opposed to simply the functional traits related to invasion and establishment (e.g., introduced species) or persistence and extinction risk (e.g., native species). When necessary, we standardised and simplified functional traits to ensure that the data were comparable across taxa and study areas (Supplementary Table 3 for more detailed information).

In addition, we analysed the community-level shifts in taxon-specific traits to account for the idiosyncrasies of each group (further details of these traits are given in Supplementary Tables 3–9). We treated species data as presence/absence since abundance information was not available for all plots.

## Urban environment characterisation

We quantified the landscape context for each plot using data from the Global Human Settlement (GHS) images analytics framework (http://ghsl.jrc.ec.europa.eu/ghs_bu_s1.php) and the Global Forest Change database[59]. These data estimate urban extents during 2016 and forest cover during the period 2000 to 2019, thus providing a reasonable estimate of land cover, as the time ranges overlap with that of the selected studies. We included the forest cover to provide an alternative landscape to the built urban land cover, in recognition that vegetation cover can be important in driving species distributions, yet different types of vegetation offer different potential resources and habitats. We recognise that for cities in more arid landscapes, forest may not reflect the natural vegetation communities, but we consider it to still be a useful landscape type given the emphasis of urban forest strategies on increasing tree canopy cover. We calculated the percent cover and level of aggregation of urban and forest land cover within a radius of 100 m

and 500 m centred on each plot for all taxa except birds, for which we use a 1000 m radius centred on each eBird checklist. We calculated the percent urban land cover around a site as the percent cover of 30 m x 30 m cells dominated by urban features (including all built-up features) using GHS. We calculated the percent forest land cover around a site with the same method, using the Global Forest Change database. To account for landscape configuration, we calculated an aggregation index[60], which is defined as the ratio of "actual shared edges" versus "maximal possible shared edges" of the 30 m x 30 m cells. Because map units do not affect the calculation, the aggregation index can be compared among classes from the same or different landscapes and even the same landscape under different buffer sizes.

We included latitude and climate data in our analyses since the composition of functional traits have been shown to vary with latitude and climate[61,62]. Latitude was based on the geographic coordinate of the sampling plot. The main statistical trends in climatic conditions were characterised using the 19 Bioclim variables of the CHELSA database[63], which provides information about biologically relevant aspects of climate for a period ranging from 1979 to 2013. We reduced the dimensionality of this dataset to limit the number of climate variables and avoid their correlations. Specifically, we ran a PCA with 100000 randomly sampled cells. We then projected the remaining cells onto the PCA. The first four PCA axes represented the main statistical trends in climate, that is, gradients in mean temperature (PC1), diurnal range (PC2), temperature seasonality (PC3) and precipitation seasonality (PC4). Altogether, these four axes accounted for ~89% of the global variation in climate (see also Supplementary Table 10) and were selected for use in the subsequent analyses.

### Functional composition of animal communities

We assessed the functional composition of the species assemblage of each taxonomic group separately. This was done by calculating the community-level mean values of each trait in each plot for each taxon or, in the case of categorical traits, the proportion of species in each category. We also calculated 10 indices capturing complementary aspects of functional trait variation: functional dispersion, functional richness, and functional evenness. To do so, we first imputed missing trait values using the K-nearest neighbours (function "preProcess" in the R package "caret"). We then calculated the Gower functional distance among species based on centered and scaled trait values. We computed a Principal Coordinates Analysis (PcoA) using the resulting functional distance matrices. The quality of trait spaces was evaluated as the absolute deviation between trait-based distance and distance in the PcoA-based space[64]. The number of axes producing the lower deviation was retained for further analyses. In the case of amphibian and reptile, many sites had 3 species or less. As a result, we retained 2 PcoA axes to be able to compute functional diversity metrics for a maximum number of sites. This represented a fair compromise between the quality of the trait space (third-best option for amphibians and second-best option for reptiles) and the number of sites to be included in the analyses. Overall, we included between 48% and 61% of the total variation in trait data in our analyses (Amphibian = 48% over 2 PcoA axes, Bat = 59% over three PcoA axes, Bee = 60% over three PcoA axes, Bird = 61% over three PcoA axes, Carabid beetles = 61% over three PcoA axes, Reptile = 53 % over two PcoA axes). We finally computed FDis, FRic, and FEve using the function alpha.fd.multidim of the R package "mFD"[65]. In addition, we computed the functional alpha diversity applied to the distance between species[66] (function "alpha.fd.hill" of the R package "mFD"), the Rao functional dispersion index[67]; the functional dispersion index based on the framework of Laliberté & Legendre[68] (function "dbFD" of the R package "FD"), the "TOP" functional richness[69]; the "TED" functional evenness[69]; and "Fever" functional evenness[70]. An example script for the calculations of these metrics is provided in a repository at https://gitlab.com/urbionet/Trait_urban_syndromes.

Since we specifically focus on functional diversity, we selected, for each aspect, the index showing the lowest correlation to species richness across all taxonomic groups (Supplementary Fig. 4, Correlations of diversity metrics across taxonomic groups). We retained the functional dispersion (FDis), functional richness (FRic), and functional evenness (FEve) indices calculated using the alpha.fd.multidim function in the R package "mFD"[65]. Functional dispersion (FDis) measures the mean distance of individual species to the centroid of all species in multidimensional trait space[68]. A decrease in FDis shows a lower dispersion of species in trait space. FDis captures aspects of both functional richness and functional evenness. Functional richness is the amount of functional niche space occupied by species within a community[68] and was calculated using the revised FRic index[65]. A decrease in FRic values suggests a decrease in the amount of functional trait space occupied by a community. Functional evenness measures how evenly species are distributed within the trait space (FEve index[71]). A decrease in FEve shows that species are less evenly distributed in trait space compared to the maximum possible (i.e., evenness = 1).

### Effect of urbanisation on faunal community functional composition

We analysed the global effect of urban land cover on the functional community composition of each taxon while controlling for the effects of forest land cover, climatic region and latitude (see Supplementary Fig. 5 for more information about the correlations among predictors). To do so, we built various models using the random forests algorithm[72]. The random forest algorithm excels at extracting patterns from complex datasets and is becoming more common in ecological studies. This approach being nonparametric, the data need not come from a specific distribution (e.g., Gaussian) and can contain collinear variables[73]. Also, random forests can deal with model selection uncertainty because predictions are based on a consensus of many models and not just a single model selected with some measure of goodness of fit. Specifically, we used the different community functional metrics as response variables, and climate PCA axes, latitude, and the percent and aggregation of urban and forest land cover as explanatory variables. Because of the observed autocorrelation in model residuals, we added spatial covariates as explanatory variables to the models. As spatial covariates, we used positive Moran's Eigenvector Maps of a distance matrix among sites (dbMEM)[74]. Relevant dbMEM were selected using a forward selection procedure based on the residuals of models computed without spatial covariates. The random forest algorithm was trained on 75% of the data and evaluated on the remaining 25%. Model training and parameter tuning were done using 2 different cross-validation strategies: 3 times 3-fold stratified CV and 30-fold spatial CV. In stratified CV, the partition is stratified according to the response variable in order to balance the class distributions within the splits (function "createDataPartition" in the R package "caret"). In spatial CV, we created 30 spatial folds for cross validation (function "CreateSpacetimeFolds" in the R package "CAST") in order to maximise the spatial transferability of model results and avoid potential overfitting. Parameter tuning used 10 random values of the number of variables to be sampled at each split time. The best model was chosen based on RMSE, MAE, and $R^2$ measured on the trained dataset. The performances of the selected model were further evaluated on the test dataset using the same metrics which we reported in Table 1. Spatial autocorrelation in model residuals was examined using Mantel correlograms (function "correlog" in the R package "vegan"). Potential overfitting was double-checked by comparing the model evaluation metrics among the train and test sets. We retained the models based on the spatial cross-validation procedure and including spatial covariates because they showed the overall best performances and the lowest potential overfitting and spatial autocorrelation of residuals.

To assess the importance of global drivers of changes in urban community functional composition, we estimated the importance of

each explanatory variable using the residual sum of squares (RSS) from random forests models. This allowed us to assess the importance of urbanisation variables amid the influence of biogeographic and macroecological processes and determine which of latitude, climatic regions, and the percent and spatial aggregation of urban land cover induce stronger changes in community functional composition.

To assess the changes in functional community composition metrics while limiting the influence of other descriptors, we used partial dependence plots (PDP)[75,76]. Partial dependence plots are especially useful for visualising the relationships discovered by complex machine learning algorithms such as random forests. PDPs help visualise the relationship between a subset of the features and the response while accounting for the average effect of the other predictors in the model (Fig. 2).

All statistical analyses were performed in R version 4.0.3, while R version 3.6.2 was used to compile the individual datasets into a consolidated dataset for each taxon[77].

### Inclusion and ethics statement

This project intentionally sought to include individuals representing a diversity of genders, career stage (university students to professors), race, cultural and linguistic diversity, and geographic distribution across countries in both the global north and global south. Criteria for authorship were communicated consistently, with taxa-coordinators assisting with translation where required. All participants who met the criteria have been included as co-authors, while those who made a partial contribution have all been included in the acknowledgements. This work was coordinated by A.K.H. from the unceded lands of the Wurundjeri Woi Wurrung, Bunurong, Dja Dja Wurrung and Wadawurrung peoples.

### Reporting summary

Further information on research design is available in the Nature Portfolio Reporting Summary linked to this article.

## Data availability

All data used in this study has been deposited in a Zenodo open data repository at https://doi.org/10.5281/zenodo.7866249. This data is publicly available.

## Code availability

All the code used in the analyses is open source and available in various R packages. A compiled version of the full code used for analysis is provided in a repository at https://gitlab.com/urbionet/Trait_urban_syndromes.

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

## Acknowledgements

This research was conducted as part of the Urban Biodiversity Research Coordination Network (UrBioNet) funded by the National Science Foundation (NSF RCN: DEB 1354676/1355151). We initiated this project as part of the Workshop Group "Patterns, Drivers and Traits" of the Urban Biodiversity Research Coordination Network

(UrBioNet, https://sites.rutgers.edu/urbionet/dsg/). We would like to thank Madhusan Katti, Christopher Trisos, and Julie Goodness for helping to conceptualise the study at the New Jersey workshop; Eliana Geretz and Carmela M. Buono for assistance with initial data compilation; Laurence Packer, Michael Batley, Stuart Roberts for bee trait expertise; and Béla Tóthmérész, Cecilia Tobar-Suárez, Etienne Normandin, Gary Luck, Lisa Smallbone, Maryna Kyrychenko-Babko, Rebecca Acosta, Yocoyani Meza-Parral for providing data included in this study. C.G.T. was also supported by an Australian Research Council Discovery Early Career Researcher Fellowship (DE200101226).

## Author contributions

The contributions of all of the people who contributed to this project are stated in the Author's CRediT statement, or in the Acknowledgements section of this paper. Criteria for authorship followed the Authorship Policy for UrBioNet based on Weltzin et al. (2006)[55] and required a contribution both leading up to the analysis, and post-analysis. The process we followed for managing contributions to this project are described in the methods. The initial project was conceptualised by: A.B.S., A.H-M., A.K.H., C.A.L., C.H.N., CGT, D.J.K., E.S.C., F.A.L., I.M-F., J.S.M., K.J., M.F.J.A., M.M., M.R.P., N.S.G.W., S.K.; the Methodology was developed by: A.B.S., A.H-M., A.K.H., B.F., C.A.L., C.H.N., CGT, D.J.K., E.S.C., F.A.L., I.M-F., J.S.M., K.J., M.F.J.A., M.M., M.R.P., N.S.G.W., S.K.; Data Curation of the individual datasets into a single taxon level dataset and was undertaken by the Taxa Coordinators: A.B.S., A.H-M., A.K.H., B.F., C.A.L., CGT, D.J.K., F.A.L., I.M-F., J.S.M., K.J., M.M., M.R.P., N.S.G.W., S.K., T.D.S.; Formal Analysis of the datasets: B.F.; Funding Acquisition led by: C.H.N., M.F.J.A.; Investigation performed by: A.B.S., A.K.H., B.F. and F.A.L.; Overall Project Administration performed by: A.K.H.; Supervision of project shared between: A.K.H., B.F., D.J.K. and M.M.; individual data sets (Resources) provided by: A.A.A., A.H-M., A.J.H., A.M.G., A.R., A.S., A.V., C.R-B., CGT, C.V., D.J.K., D.L., D.M.L., D.O., D.T., E.B., E.P., E.S.C., G.L., I.M-F., J.C., J.C.M., J.C-O., J.P.S., J.S., J.S.M., J.T.S., K.B., K.C.M., K.C.R.B., K.J., K.K-M., L.F.A., L.O.A., L.S., M.I.H-M., M.K.O., M.M., M.Z., N.S.G.W., R.K.T., R.T., S.A.G., S.J.H., S.K., T.D.S., T.L., T.M., T.S., W.U. and Z.E.; Software: B.F. and F.A.L.; Visualisation and production of figures undertaken by: A.K.H., B.F., D.J.K. and M.M.; Writing- original drafts and subsequent revisions: A.K.H., B.F., D.J.K. and M.M.; Writing- Review & editing: A.A.A., A.B.S., A.H-M., A.J.H., A.K.H., A.M.G., A.R., A.S., A.V., B.F., C.A.L., C.H.N., C.R-B., CGT, C.V., D.J.K., D.L., D.M.L., D.O., D.T., E.B., E.P., E.S.C., F.A.L., G.L., I.M-F., J.C., J.C.M., J.C-O., J.P.S., J.S., J.S.M., J.T.S., K.B., K.C.M., K.C.R.B., K.J., K.K-M., L.F.A., L.O.A., L.S., M.F.J.A., M.I.H-M., M.K.O., M.M., M.R.P., M.Z., N.S.G.W., R.K.T., R.T., S.A.G., S.J.H., S.K., T.D.S., T.L., T.M., T.S., W.U. and Z.E. All authors reviewed the manuscript and approved it for submission.

## Competing interests

The authors declare no competing interests.

## Additional information

Amy K. Hahs [1,62] ✉, Bertrand Fournier [2,62] ✉, Myla F. J. Aronson [3], Charles H. Nilon[4], Adriana Herrera-Montes[5], Allyson B. Salisbury[6], Caragh G. Threlfall[7,8], Christine C. Rega-Brodsky [9], Christopher A. Lepczyk [10], Frank A. La Sorte [11], Ian MacGregor-Fors[12], J. Scott MacIvor[13], Kirsten Jung[14], Max R. Piana [15], Nicholas S. G. Williams [1], Sonja Knapp [16,17,18], Alan Vergnes[19], Aldemar A. Acevedo [20], Alison M. Gainsbury [21], Ana Rainho [22], Andrew J. Hamer [23], Assaf Shwartz[24], Christian C. Voigt[25], Daniel Lewanzik [25], David M. Lowenstein[26], David O'Brien[27], Desiree Tommasi[28], Eduardo Pineda[29], Ela Sita Carpenter[30], Elena Belskaya[31], Gábor L. Lövei [32,33], James C. Makinson[34], Joanna L. Coleman [35], Jon P. Sadler[36], Jordan Shroyer[4], Julie Teresa Shapiro[37], Katherine C. R. Baldock [38,39,40], Kelly Ksiazek-Mikenas[41], Kevin C. Matteson[42], Kyle Barrett[43], Lizette Siles [44], Luis F. Aguirre [45], Luis Orlando Armesto [46], Marcin Zalewski[47], Maria Isabel Herrera-Montes[48], Martin K. Obrist [49], Rebecca K. Tonietto[50], Sara A. Gagné[51], Sarah J. Hinners [52], Tanya Latty[53], Thilina D. Surasinghe [54], Thomas Sattler [55], Tibor Magura [33,56], Werner Ulrich[57], Zoltan Elek[58], Jennifer Castañeda-Oviedo[59], Ricardo Torrado[60], D. Johan Kotze [12,62] ✉ & Marco Moretti [61,62] ✉

¹School of Agriculture, Food and Ecosystem Sciences, The University of Melbourne, Burnley Campus 500 Yarra Blvd, Richmond 3121 VIC, Australia. ²Institute of Environmental Science and Geography, University of Potsdam, Karl-Liebknecht-Str. 24-25, 14476 Potsdam, Germany. ³Department of Ecology, Evolution and Natural Resources, Rutgers, The State University of New Jersey, New Brunswick, NJ 08816, USA. ⁴School of Natural Resources, University of Missouri,

Columbia, MO 65211, USA. [5]Department of Environmental Science, College of Natural Sciences, University of Puerto Rico, San Juan, Puerto Rico. [6]The Morton Arboretum, 4100 Illinois Route 53, Lisle, IL 60532, USA. [7]School of Life and Environmental Sciences, The University of Sydney, Sydney, NSW 2006, Australia. [8]School of Natural Sciences, Macquarie University, Sydney, NSW 2109, Australia. [9]School of Science and Mathematics, Pittsburg State University, Pittsburg, KS 66762, USA. [10]School of Forestry, Wildlife and Environment, Auburn University, Auburn, AL 36849, USA. [11]Cornell Lab of Ornithology, Cornell University, Ithaca, NY 14850, USA. [12]Faculty of Biological and Environmental Sciences, Ecosystems and Environment Research Programme, University of Helsinki, Niemenkatu 73, FI-15140 Lahti, Finland. [13]Department of Biological Sciences, University of Toronto Scarborough, 1265 Military Trail, Toronto M1C 1A4, Canada. [14]Institute of Evolutionary Ecology and Conservation Genomics, Ulm University, Albert-Einstein-Allee 11, 89069 Ulm, Germany. [15]USDA Forest Service, Northern Research Station, Amherst, MA 01002, USA. [16]Helmholtz Centre for Environmental Research – UFZ, Department of Community Ecology, Theodor-Lieser-Str. 4, 06120 Halle (Saale), Germany. [17]German Centre for Integrative Biodiversity Research (iDiv) Halle-Jena-Leipzig, Puschstraße 4, 04103 Leipzig, Germany. [18]Technische Universität Berlin, Department of Plant Ecology, Rothenburgstraße 12, 12165 Berlin, Germany. [19]CEFE, Univ Montpellier, CNRS, EPHE, IRD, Univ Paul Valéry Montpellier 3, Montpellier, France. [20]Departamento de Ciencias Ecológicas, Facultad de Ciencias, Laboratorio de Genética y Evolución, Universidad de Chile, Las Palmeras 3425, Ñuñoa, Santiago, Chile. [21]University of South Florida, St. Petersburg Campus, Department of Integrative Biology, St. Petersburg, FL 33701, USA. [22]cE3c – Centre for Ecology, Evolution and Environmental Changes at the Dept. of Animal Biology, Faculty of Sciences, Univ. of Lisbon, Lisboa, Portugal. [23]Institute of Aquatic Ecology, Centre for Ecological Research, Karolina u. 29, 1113 Budapest, Hungary. [24]Faculty of Architecture and Town Planning, Technion – Israel Institute of Technology, Haifa 32000, Israel. [25]Dept. of Evolutionary Ecology, Leibniz Institute for Zoo and Wildlife Research, Alfred-Kowalke-Str. 17, 10315 Berlin, Germany. [26]Michigan State University Extension, Macomb County, 21885 Dunham Rd - Suite 12, Clinton Twp, MI 48036, USA. [27]Scottish Natural Heritage (NatureScot), Great Glen House, Inverness IV3 8NW, UK. [28]Institute of Marine Sciences, University of California Santa Cruz, Santa Cruz, CA 95064, USA. [29]Red de Biología y Conservación de Vertebrados. Instituto de Ecología, A.C. Carretera Antigua a Coatepec 351, Xalapa 91073, Mexico. [30]U.S. Fish and Wildlife Service, Chesapeake Bay Field Office, 177 Admiral Cochrane Dr, Annapolis, MD 21401, USA. [31]Institute of Plant and Animal Ecology, Ural Branch, Russian Academy of Sciences, Eighth March Street 202, Yekaterinburg 620144, Russia. [32]Department of Agroecology, Aarhus University, Flakkebjerg Research Centre, DK-4200 Slagelse, Denmark. [33]ELKH-DE Anthropocene Ecology Research Group, University of Debrecen, H-4032 Debrecen, Egyetem square 1, Hungary. [34]Hawkesbury Institute for the Environment, Western Sydney University, Locked Bag 1797, Penrith, NSW 2751, Australia. [35]Queens College at the City University of New York, Flushing, NY, USA. [36]School of Geography, Earth and Environmental Sciences, University of Birmingham, Edgbaston, Birmingham B15 2TT, UK. [37]University of Lyon, French Agency for Food, Environmental and Occupational Health & Safety (ANSES), Laboratory of Lyon, 31 Avenue Tony Garnier, 69364 Lyon Cedex 07, France. [38]Department of Geography and Environmental Sciences, Northumbria University, Newcastle upon Tyne, UK. [39]School of Biological Sciences, University of Bristol, Bristol, UK. [40]Cabot Institute, University of Bristol, Bristol, UK. [41]Department of Biology, Elmhurst University, Elmhurst, IL 60126, USA. [42]Department of Biology/Project Dragonfly, Miami University, Oxford, OH, USA. [43]Department of Forestry and Environmental Conservation, Clemson University, 261 Lehotsky Hall, Clemson, SC 29631, USA. [44]Área de Mastozoología, Museo de Historia Natural Alcide d'Orbigny, Avenida Potosí 1458, Cochabamba, Cochabamba, Bolivia. [45]Centro de Biodiversidad y Genética, Universidad Mayor de San Simón, c Sucre, frente Parque La Torre s/n, Cochabamba, Bolivia. [46]Tecnoacademia, CEDRUM, Servicio Nacional de Aprendizaje (SENA), Cúcuta, Colombia. [47]Museum and Institute of Zoology of the Polish Academy of Sciences, Wilcza 64, Warsaw 00-679, Poland. [48]Grupo de Ecologia Animal, Universidad del Valle, Cali, Colombia. [49]Swiss Federal Institute for Forest, Snow and Landscape Research WSL, Biodiversity and Conservation Biology, CH-8903 Birmensdorf, Switzerland. [50]Department of Natural Sciences, University of Michigan-Flint, 303 E Kearsley St., Flint, MI 48502, USA. [51]University of North Carolina at Charlotte, 9201 University City Blvd., Charlotte, NC 28223, USA. [52]Department of City and Metropolitan Planning, University of Utah, Salt Lake City, UT, USA. [53]Sydney Institute of Agriculture, School of Life and Environmental Sciences, University of Sydney, Sydney, Australia. [54]Department of Biological Sciences, Bridgewater State University, Bridgewater, MA 02325, USA. [55]Swiss Ornithological Institute, Seerose 1, CH-6204 Sempach, Switzerland. [56]Department of Ecology, Faculty of Science and Technology, University of Debrecen, H-4032 Debrecen, Egyetem square 1., Hungary. [57]Department of Ecology and Biogeography, Nicolaus Copernicus University, Lwowska 1, 87-100 Torun, Poland. [58]Centre for Agricultural Research, Plant Protection Institute, Eötvös Loránd Research Network, Herman Ottó út 15, Budapest 1022, Hungary. [59]Grupo de Investigación en Ecología y Biogeografía, Universidad de Pamplona, Pamplona, Colombia. [60]Secretaría de Educación del Municipio de Cúcuta, Cúcuta, Colombia. [61]Swiss Federal Research Institute WSL, Biodiversity and Conservation Biology, Zürcherstrasse 111, 8903 Birmensdorf, Switzerland. [62]These authors contributed equally: Amy K. Hahs, Bertrand Fournier, D. Johan Kotze, Marco Moretti. ✉e-mail: amy.hahs@unimelb.edu.au; bertrand.fournier@uni-potsdam.de; johan.kotze@helsinki.fi; marco.moretti@wsl.ch

