## [Peer Review File · Nature Communications]

Urbanisation generates multiple trait syndromes for terrestrial animal taxa worldwideEditorial Note: This manuscript has been previously reviewed at another journal that is not operating a transparent peer review scheme. This document only contains reviewer comments and rebuttal letters for versions considered at *Nature Communications*.

REVIEWERS' COMMENTS

Reviewer #2 (Remarks to the Author):

The manuscript "Urbanisation generates multiple trait syndromes for terrestrial taxa worldwide" is an interesting contribution to the arena of urban ecology. As previously highlighted, this topic is rather important, because understanding the biological impacts of urbanization on wildlife is still debated in the scientific community, and a global study, merging the results of different taxonomic groups provides useful information for conservation planning.

In this new version of the manuscript, the authors correctly addressed my main concerns raised during the review process previously initiated in Nature.

The main concern I raised previously, was related to the novelty of the study, that is - in my opinion - partially compromised by some studies published previously. However, as the authors pointed out now, I agree that they are offering a novel contribution through the global compilation of datasets that simultaneously consider multiple but shared traits from multiple taxa investigated at the site scale. Then, I think the study deserves to be published.

Reviewer #3 (Remarks to the Author):

I have commented on the earlier version of the manuscript entitled "Urbanisation generates multiple trait syndromes for terrestrial taxa worldwide" in the earlier version submitted to the journal Nature. In general, this is an excellent study based on a robust dataset showing the effects of urbanization on the functional diversity of several taxonomic groups of animals. The study is novel, the data are representative, and the methods are appropriate. Compared to the previous version, the manuscript has been dramatically improved. The authors have resolved several of my comments regarding missing taxonomic groups, spatial autocorrelation, and climatic data. I have only minor comments on the new version:

Please explain what you mean by the term "urbanization syndrome" used in the abstract. I see a possible misunderstanding here since your paper identifies four different syndromes. Please be more specific.

Ecosystem services: The term is used in the introduction (l. 191, but further in the text, the term is not used). Perhaps it is not necessary to include a term that is no longer used in the study.

The discussion chapter should start with some general statements related to your hypothesis. Please, add.

l. 132 I am afraid you can not study local extinction with your dataset

l. 303-304 repetition of the text in the results

Reviewer #4 (Remarks to the Author):

This manuscript is much improved compared to the first version I reviewed. The authors have gone to substantial lengths to address my comments and I am broadly satisfied. I think the resulting paper is much improved and worthy of publication.

Minor query: the authors have taken on board the point about model overfitting and conducted tests of predictive capacity on test and training datasets (which is excellent). I found it slightly odd to see the reported % variance explained in table 1 where some values are very high (e.g. up to 80% of the variation in data explained by model). Wouldn't it be better to present to % variation explain in the test dataset rather than training dataset? This is a more genuine reflection of the predictive capacity of these models?

Minor comment: The analysis into the four urban trait syndromes is interesting but arguably speculative, where those groups are defined post hoc in an attempt to explain the variable responses across species? Therefore, I would suggest weakening statements such as this one in the conclusion "Overall, our results suggest that resource distribution and abundance are filtering taxa into one of four urban trait syndromes: mobile generalists, mobile specialists (nomads), central place foragers and site scale specialists" You don't have hypotheses or conduct statistical tests about whether species fit into these syndromes, so I don't think you can state that species are being filtered into only these four categories.

REVIEWERS' COMMENTS

Reviewer #2 (Remarks to the Author):

The manuscript "Urbanisation generates multiple trait syndromes for terrestrial taxa worldwide" is an interesting contribution to the arena of urban ecology. As previously highlighted, this topic is rather important, because understanding the biological impacts of urbanization on wildlife is still debated in the scientific community, and a global study, merging the results of different taxonomic groups provides useful information for conservation planning.

In this new version of the manuscript, the authors correctly addressed my main concerns raised during the review process previously initiated in Nature.

The main concern I raised previously, was related to the novelty of the study, that is - in my opinion - partially compromised by some studies published previously. However, as the authors pointed out now, I agree that they are offering a novel contribution through the global compilation of datasets that simultaneously consider multiple but shared traits from multiple taxa investigated at the site scale.

Then, I think the study deserves to be published.

RESPONSE R2 COMMENT 1: Thank you for your positive response.

R1 COMMENT 1:

Reviewer #3 (Remarks to the Author):

I have commented on the earlier version of the manuscript entitled "Urbanisation generates multiple trait syndromes for terrestrial taxa worldwide" in the earlier version submitted to the journal Nature. In general, this is an excellent study based on a robust dataset showing the effects of urbanization on the functional diversity of several taxonomic groups of animals. The study is novel, the data are representative, and the methods are appropriate. Compared to the previous version, the manuscript has been dramatically improved. The authors have resolved several of my comments regarding missing taxonomic groups, spatial autocorrelation, and climatic data. I have only minor comments on the new version:

RESPONSE R3 COMMENT 1: Thank you for your positive feedback.

Please explain what you mean by the term "urbanization syndrome" used in the abstract. I see a possible misunderstanding here since your paper identifies four different syndromes. Please be more specific.

RESPONSE R3 COMMENT 2: Thanks for picking up on that. We agree that the "global urban syndrome" was not explained clearly in the abstract. We revised the sentence as follows: "This study suggests that urbanisation results in four general urban trait syndromes (mobile generalists, site specialists, central place foragers, and mobile specialists), with resources

associated with reproduction and diet likely to be driving patterns in traits associated with mobility and body size.”

Ecosystem services: The term is used in the introduction (l. 191, but further in the text, the term is not used). Perhaps it is not necessary to include a term that is no longer used in the study.

RESPONSE R3 COMMENT 3: We agree and removed the term “ecosystem services” from the sentence. Note that the mention of “ecosystem services” in the first sentence of the introduction was kept.

The discussion chapter should start with some general statements related to your hypothesis. Please, add.

RESPONSE R3 COMMENT 4: We added a few sentences at the beginning of the discussion to address this point.

l. 132 I am afraid you cannot study local extinction with your dataset

RESPONSE R3 COMMENT 5: We rephrased the sentence as follows: “Yet, urbanisation leads to the loss of habitats and, potentially, species and functional groups.”

l. 303-304 repetition of the text in the results

RESPONSE R3 COMMENT 6: We rephrased this sentence as well as the one about amphibians and reptiles in the same paragraph.

Reviewer #4 (Remarks to the Author):

This manuscript is much improved compared to the first version I reviewed. The authors have gone to substantial lengths to address my comments and I am broadly satisfied. I think the resulting paper is much improved and worthy of publication.

RESPONSE R4 COMMENT 1: Thank you for your positive feedback.

Minor query: the authors have taken on board the point about model overfitting and conducted tests of predictive capacity on test and training datasets (which is excellent). I found it slightly odd to see the reported % variance explained in table 1 where some values are very high (e.g. up to 80% of the variation in data explained by model). Wouldn't it be better to present to % variation explain in the test dataset rather than training dataset? This is a more genuine reflection of the predictive capacity of these models?

RESPONSE R4 COMMENT 2: We indeed agree with this comment. However, Table 1 already reports model performances on the test datasets, not on the training dataset. We have made that clearer in the caption of the table and in the methods. In addition, we replaced the term “% explain” by “ $R^2_{\text{obs-pred}}$ ” in the Results section to avoid potential confusions about the performances of our models.

Minor comment: The analysis into the four urban trait syndromes is interesting but arguably

speculative, where those groups are defined post hoc in an attempt to explain the variable responses across species? Therefore, I would suggest weakening statements such as this one in the conclusion "Overall, our results suggest that resource distribution and abundance are filtering taxa into one of four urban trait syndromes: mobile generalists, mobile specialists (nomads), central place foragers and site scale specialists" You don't have hypotheses or conduct statistical tests about whether species fit into these syndromes, so I don't think you can state that species are being filtered into only these four categories.

RESPONSE R4 COMMENT 3: That is a fair point. We rephrased the sentence in the conclusion as follows:

“Overall, our results suggest that resource distribution and abundance appear to filter taxa into one of four urban trait syndromes: mobile generalists, mobile specialists (nomads), central place foragers and site scale specialists. ... Future research would be needed to untangle these relationships further.”